# Differential Expression Analysis Reveals Possible New Quaternary Ammonium Compound Resistance Gene in Highly Resistant *Serratia* sp. HRI

**DOI:** 10.3390/microorganisms12091891

**Published:** 2024-09-13

**Authors:** Samantha McCarlie, Charlotte Boucher-van Jaarsveld, Robert Bragg

**Affiliations:** Department of Microbiology and Biochemistry, University of the Free State, Bloemfontein 9301, South Africa; samanthamccarlie@gmail.com (S.M.); boucherce@gmail.com (C.B.-v.J.)

**Keywords:** biocides, antimicrobial resistance, MFS efflux, RNA-Seq, qPCR

## Abstract

During the COVID-19 pandemic, the surge in disinfectant use emphasised their pivotal role in infection control. While the majority of antimicrobial resistance research focuses on antibiotics, resistance to biocides, which are present in disinfectants and sanitisers, is escalating. *Serratia* sp. HRI is a highly resistant isolate, and through the study of this organism, the molecular mechanisms of resistance may be uncovered. *Serratia* sp. HRI was treated with the disinfectant benzalkonium chloride in preparation for RNA sequencing. Through mining of the RNA-Seq differential expression data, an uncharacterised Major Facilitator Superfamily (MFS) efflux pump gene was found to be up-regulated at least four-fold at four different time points of exposure. Real-time PCR revealed this uncharacterised MFS efflux gene was up-regulated after exposure to benzalkonium chloride and two additional disinfectants, didecyldimethylammonium chloride (DDAC) and Virukill^TM^. Additionally, expression of this gene was found to be higher at 20 min versus 90 min of exposure, indicating that the up-regulation of this gene is an initial response to biocide treatment that decreases over time. This suggests that MFS efflux pumps may be an initial survival mechanism for microorganisms, allowing time for longer-term resistance mechanisms. This work puts forward a novel biocide resistance gene that could have a major impact on biocide susceptibility and resistance.

## 1. Introduction

Antimicrobial resistance is a well-known issue, although the vast majority of research focuses on resistance to antibiotics. A lesser-researched form of antimicrobial resistance is resistance to biocides, which form the active ingredients in disinfectants, antiseptics, and hand sanitisers. During the COVID-19 pandemic, our reliance on biocides was highlighted through extensive use of disinfectants and sanitisers for infection prevention. These products are not only crucial in healthcare for infection control but are important in agriculture for disease protection and in the food and beverage industry to maintain hygiene protocols, thus safeguarding food security worldwide [1].

During the COVID-19 pandemic, the United States Environmental Protection Agency (EPA) published a list of approved disinfectants for use against SARS-CoV-2 [2]. Disinfectant products containing quaternary ammonium compounds (QACs) as the active ingredient were the majority of disinfectants approved [2]. Within this group, benzalkonium chloride (BC) and didecyldimethylammonium chloride (DDAC) were the most common active ingredients [2]. Alarmingly, overuse and improper applications of disinfectants during the pandemic increased the selective pressure on microorganisms and has accelerated the development of reduced susceptibility and resistance to biocides [3,4,5,6]. In addition, this resistance is not specific, as research has shown that if a microorganism develops resistance to biocides, antibiotic resistance can develop simultaneously without prior exposure [7,8].

New advances in sequencing technology and data analysis, coupled with broader databases, have revealed the impact of molecular mechanisms of antimicrobial resistance [9,10,11]. This has allowed for the discovery of novel resistance genes and mechanisms, and for the surveillance of antimicrobial determinants [11,12,13,14]. Together, this allows for further insight into the epidemiology of resistance and elucidation of how resistant populations develop. One such microorganism is *Serratia* sp. HRI, an isolate with high resistance capabilities to several active ingredients used in disinfectants and other biocides [15,16]. Genome sequence analysis revealed *Serratia marcescens* to be the closest related species [16], which is a notorious pathogen responsible for a number of nosocomial outbreaks in neonatal and paediatric facilities [17,18,19,20]. The study of this isolate is a unique opportunity that may reveal novel molecular mechanisms that confer the highly resistant phenotype. Three different generations of QAC products were used in this study, as these products are used globally in medical facilities and agriculture for infection control. This work used transcriptomics in the form of RNA-Seq to screen for exceedingly differentially expressed genes after exposure to BC at various time points. After mining the transcriptomic data, lists of novel targets for QAC-resistance genes were generated. One gene was singled out in this work and real-time PCR was performed to further analyse the differential expression of this gene after exposure to two additional QAC disinfectants: DDAC and Virukill^TM^. This work has generated lists of possible novel QAC-resistance genes for further study and confirmed the significant differential expression and role played by MFS-efflux pumps in QAC resistance.

## 2. Materials and Methods

### 2.1. Bacterial Strains

This study focused on the highly resistant isolate *Serratia* sp. HRI [16] obtained from the culture collection of the University of the Free State. This strain was part of a batch of mixed environmental samples and was subsequently found in its pure form contaminating a bottle of QAC disinfectant, where it was isolated [16].

### 2.2. Disinfectants and Conditions

Benzalkonium chloride (BC) was obtained from Sigma-Aldrich (St. Louis, MO, USA) (≥50% in water), while didecyldimethylammonium chloride (DDAC) (80% UNIQUAT) and Virukill™ (DDAC 120 g/L) were both obtained from ICA International Chemicals (Plankenbrug, South Africa). The minimal inhibitory concentration (MIC) levels of these disinfectants for *Serratia* sp. were previously reported [16], and in this study, all the disinfectants were used at sub-MIC levels as follows: BC at a sub-MIC level of 9 800 mg/L, DDAC at 76.62 mg/L, and Virukill™ at 1 200 mg/L.

### 2.3. Bacterial Cultivation

The *Serratia* sp. HRI was prepared as described in previous growth kinetics studies [15]. After 60 min, a sub-MIC level of benzalkonium chloride was added to the inoculum (except the untreated control samples) and a contact time of 10, 20, 30, or 90 min was allowed, as depicted in Table 1, before the cell pellets were immediately treated with Invitrogen RNAlater Stabilization Solution to store and stabilise the RNA within the cells before RNA extraction commenced (Invitrogen, Carlsbad, CA, USA).

### 2.4. RNA Extraction and Quality Control

RNA extraction was performed on fresh cell pellets using the RNeasy mini kit (Qiagen, Hilden, Germany) according to the manufacturer’s instructions. An additional on-column DNAse digestion was performed to eliminate possible genomic DNA (gDNA) contamination using the ZymoResearch DNase I set as per the manufacturer’s instructions (ZymoResearch, Irvine, CA, USA), and the DNA digestion was confirmed by agarose gel electrophoresis. The total RNA integrity and purity assessments were performed using the 2100 Agilent Bioanalyzer using the Agilent RNA 6000 Nano Kit (Agilent Technologies, Santa Clara, CA, USA) following the manufacturer’s protocol. All the samples had to adhere to an RIN value of ≥7.5. The purity (A260/A280 ratios) and concentrations were determined with the NanoDrop 1000 spectrophotometer and the concentration of the RNA samples was measured using the Qubit RNA BR Assay Kit (Thermo Fisher Scientific, Waltham, MA, USA). Once all the samples had passed the QC based on the RNA integrity, concentration and purity, the samples were sent to the Centre for Proteomic and Genomic Research for sequencing (Cape Town, South Africa) and were prepared as described below. RNA extracts were used for both the RNA-Seq and conversion to cDNA for real-time downstream validation.

The total RNA for all the samples was precipitated by the addition of sodium acetate (0.3 M) and 3 volumes of ice-cold absolute ethanol. The RNA was allowed to precipitate at −80 °C overnight. The RNA was pelleted by centrifugation at 13,000× *g* for 10 min. The RNA pellets were washed twice with ice-cold 70% ethanol and centrifuged at 13,000× *g* for 10 min. The ethanol was removed and the pellets were dried at room temperature and then resuspended in nuclease-free water. The RNA was evaluated for purity using the NanoDrop 8000 spectrophotometer, for quantity using the Qubit RNA BR Assay Kit, and for integrity using the Agilent Bioanalyzer RNA 6000 Nano Kit. Bacterial ribosomal RNA was depleted using the Invitrogen RiboMinus Bacteria module according to the manufacturer’s instructions, using 5 μg of each sample. The resulting ribosomal-depleted RNA was concentrated using the Invitrogen RiboMinus Concentration Module.

### 2.5. RNA-Seq and QC

The ribosomal-depleted RNA was used for the Illumina library preparation using the NEXTflex Rapid Directional RNA-Seq Kit according to the manufacturer’s instructions. The ribosomal-depleted RNA was fragmented and converted to single-stranded cDNA, followed by second-strand synthesis using dUTP. The cDNA was purified by using Agencourt AMPure XP magnetic beads, followed by adenylation and adapter ligation. The adapter-ligated cDNA was purified by using Agencourt AMPure XP magnetic beads, then treated with Uracil DNA Glycosylase to maintain the strand information, and finally, amplified by PCR for 15 cycles. The libraries were purified using Agencourt AMPure XP magnetic beads. The purified RNA-Seq libraries were quantified using the Qubit 1X dsDNA HS Assay Kit. The library size distribution for each library was determined using the Agilent TapeStation D1000 assay (D1000 ScreenTape (5067–5582) with D1000 Reagents (5067–5583). The RNA-Seq libraries were subsequently pooled in equimolar amounts for sequencing. The average fragment size for the pooled library was determined using the Agilent Bioanalyzer HS DNA and the region analysis function of the Bioanalyzer software. The concentration of adaptor-ligated DNA molecules was confirmed by qPCR using the KAPA Library Quantification Kit for Illumina Platforms on the QuantStudio 12K Flex System. A library dilution series of 1:10,000, 1:50,000, 1:250,000, and 1:1250,000 was quantified in triplicate. The size-adjusted library concentration was calculated using the KAPA Library Quantification Kit (Illumina Platforms) Data Analysis Template (v4.14). The RNA-Seq pooled libraries were denatured with 0.2 N NaOH, diluted to 1.5 pM and combined with the denatured PhiX positive control at a spike-in concentration of 1% *v*/*v* for the NGS run according to the standard Illumina protocol. The pooled library was loaded on the Illumina NextSeq 550 instrument (serial no. NB552161) and sequenced using a NextSeq 500/550 Mid Output Kit v2.5 (150 Cycles) consisting of the following: Mid Output Flow Cell Cartridge v2.5; Buffer Cartridge v2; Mid Output Reagent Cartridge v2 150 cycles; Accessory Box v2. The sequencer was programmed to perform a paired-end, single-indexed 2 × 76 cycle sequencing run (Illumina Inc., San Diego, CA, USA).

### 2.6. RNA-Seq Sample Data Analysis

The RNA-Seq read libraries were uploaded to the Bacterial and Viral Bioinformatics Resource Center (BV-BRC) and an automated RNA-Seq Analysis pipeline was used for the QC of the raw reads, trimming, alignment, assembling, and differential expression analysis. A previously published full-genome sequence of *Serratia* sp. HRI was used as a reference genome for the paired-read library input [16]. The Tuxedo pipeline was used based on the tuxedo suite of tools (i.e., Bowtie2, Cufflinks, Cuffdiff) as part of TopHat 2 [17] and the computational analysis of the bacterial RNA-Seq data according to standards set by McClure and co-workers [18]. The differential expression *p*-values were computed by q-values based on Benjamini–Hochberg correction with a false recovery rate of <1%.

### 2.7. Uncharacterised MFS-Efflux Pump

Based on the differential expression analysis and screening of the RNA-Seq results, an uncharacterised MFS efflux pump (Detailed in Table 2) was found to be up-regulated at all the time points during the BC treatment. Additionally, only one copy of this gene was found in the genome and so this gene was chosen as a target for further analysis through quantitative real-time expression studies. All the methodology and results analysis was performed following the MIQE guidelines [19]. The treatment conditions included BC treatment at 20 and 90 min and expanded to include two additional QAC biocides, DDAC and Virukill^TM^, with contact times of 20 min. The characteristics of the target gene were as follows.

### 2.8. Reverse Transcription of RNA to cDNA

For the validation and further analysis after the RNA-Seq, biological replicates of the same RNA extracts were converted to cDNA using the Applied Biosystems™ High-Capacity RNA-to-cDNA™ Kit as per the manufacturer’s instructions (Applied Biosystems, Waltham, MA, USA). Thereafter, the cDNA concentrations were measured using the 1X ssDNA Qubit Assay Kit and normalised to 10 ng/µL for each cDNA sample.

### 2.9. Quantitative Real-Time PCR Using SYBR^TM^ Green Assay

Quantitative PCR (qPCR) was carried out in technical and biological triplicates for the uncharacterised MFS efflux pump gene under each condition using the Applied Biosystems™ PowerUp™ SYBR™ Green Master Mix on an Applied Biosystems QuantStudio^TM^ 5 Real-Time PCR system according to the manufacturer’s instructions. Untreated samples were compared to four conditions: treatment with BC for 20 min (BC20), treatment with BC for 90 min (BC90), treatment with DDAC for 20 min (D), and treatment with Virukill^TM^ for 20 min (V). The standard concentrations were confirmed using the 1X ssDNA Qubit Assay Kit and the gene copy numbers were extrapolated using the Thermofisher DNA Copy Number and Dilution Calculator available at https://www.thermofisher.com/za/en/home/brands/thermo-scientific/molecular-biology/molecular-biology-learning-center/molecular-biology-resource-library/thermo-scientific-web-tools/dna-copy-number-calculator.html, accessed on 9 July 2024. 

In addition to the in silico analysis and verification using Geneious 11.1.5 software, the designed primers were tested for non-specific amplification of the *Serratia* sp. HRI cDNA and optimised based on pooled biological replicates using agarose gel electrophoresis and the Applied Biosystems Quant StudioTM 5 Real-Time PCR system. All the qPCR samples were set up and run as described below, followed by a dissociation (melt) curve. Standard ramp speeds were used with the reporter molecule SYBRTM Green and passive reference dye ROX^TM^ with no quencher.

The results were analysed using QuantStudio Design & Analysis 2 (DA2) software version 2. Based on the QC and analysis results, standard curves were constructed by DA2, together with the Cq and gene copy numbers of unknown samples relative to each standard curve generated during each run.

## 3. Results

### 3.1. Screening for Novel QAC-Resistance Genes

#### 3.1.1. Total RNA Concentration, Integrity and Purity Assessment

Total RNA extracts were tested for purity, integrity and concentration as depicted in Figure 1 and Table 3.

The total RNA extracts passed all QC with high concentrations and low levels of degradation; the purity values were found to indicate high quality for downstream analysis.

#### 3.1.2. Differential Gene Expression (DGE) Analysis

Figure 2 and Figure 3 were generated after the DGE of the samples at different time points. In Figure 2, a great deal of red bars indicate a large number of up-regulated genes, with some down-regulation indicated by green bars. When considering the overall patterns, the highest number of genes differentially expressed, compared to the control, was at 10 min and 30 min of treatment. Figure 3 indicates the changes in gene expression over time in the treated samples and suggests that some genes are differentially expressed over time during treatment; however, not many genes are affected. This indicates that most differential expression is sustained after the 10 min mark.

Due to the large numbers of differentially expressed genes, seen in Figure 2 between the control and treated samples, the results were filtered as follows: (1) a fold-change ≥ 3 and Z-score ≥ 3 (a statistical measurement of the differential expression of a gene in relation to the mean of the group). This sub-section of genes, depicted in Figure 3 and quantified in Table 4,were deemed to be the most differentially expressed and were sorted into groups based on the trends of DGE after exposure to BC previously recorded in the literature.

Table 4 indicates the genes most differentially expressed after exposure to BC at four different time points, where the majority of the genes are uncharacterised for all four time-points.

### 3.2. qPCR Analysis of Novel QAC-Resistance Target Gene: Uncharacterised MFS Efflux Pump (UMFS)

When searching for target genes for qPCR validation, an uncharacterised MFS efflux pump (RefSeq: WP_017892467.1; gf111_02590) was found to be up-regulated by at least four-fold and to have a Z-score of above 3 at all four time-points. As a result, this gene was chosen as a target for further analysis and validation using qPCR. Samples were collected after exposure to BC at 20 and 90 min as well as DDAC and Virukill^TM^ disinfectants at 20 min of contact time to elucidate more about this gene under different conditions. Template (cDNA) concentration is depicted in Table 5 for all samples after conversion from RNA. Followed by qPCR analysis parameters achieved in Table 6. 

Figure 4 indicates the successful amplification, single bands and the expected amplicon size.

The results of the real-time PCR analysis depicted in Figure 5 reveal further information about the UMFS1 target gene. An up-regulation of this gene after exposure to BC at the 20 and 90 min time points was confirmed (Table 7), together with an up-regulation in response to the DDAC and Virukill^TM^ disinfectants (Figure 5). Interestingly, the UMFS1 gene was up-regulated at a much higher level at 20 min compared to 90 min.

## 4. Discussion

*Serratia* sp. HRI is an isolate with high resistance capabilities for a wide range of antimicrobials [16]. As such, any insight into the molecular mechanisms of the QAC resistance of this isolate are valuable in the fight against antimicrobial resistance.

Data gathered from RNA-Seq of the transcriptomic response of *Serratia* sp. HRI to benzalkonium chloride shows that the differential gene expression changes over time. The main systems affected were the motility, membrane composition and transport, metabolism adaption, and stress response. The immediate (≤10 min) response was characterised by the up-regulation of genes associated with mobile genetic elements, efflux pumps and membrane transporters, biofilm maintenance, pentose phosphate pathway (PPP) metabolites and enzymes, metabolites of the tricarboxylic acid (TCA) cycle and detoxification, antibiotic and resistance genes. After 20 min of exposure, less genes and systems were differentially expressed compared to the initial 10 min readings. The differential expression was observed as a down-regulation of lipoproteins, up-regulation of oxidative and acidic stress responses, and up-regulation of pyruvate production and citrate fermentation feeding into the TCA cycle. After 30 min of exposure, the differential expression profile shifted and more genes were activated or up-regulated, specifically mobile genetic elemental genes, efflux pumps and membrane proteins, copper resistance, biofilm maintenance, TCA intermediates, PPP intermediates, antibiotic and copper resistance genes, and finally, the osmotic stress response. When compared to the reviewed literature, these trends align with an up-regulation of the stress response, efflux pumps and membrane transport, a metabolism shift due to oxidative damage and proteostasis [20]; however, in contrast to the literature, no changes in the expression of iron homeostasis genes were observed for *Serratia* sp. HRI in the short-term (up to 30 min) response to benzalkonium chloride. The long-term (characterised here as 90 min exposure) differential expression patterns in *Serratia* sp. HRI were an up-regulation of genes involved in membrane maintenance, TCA cycle metabolites, NADPH formation, copper and antibiotic resistance, together with the differential expression of mobile genetic elements. Interestingly, all the multidrug and resistance efflux pumps were most differentially expressed within the first 10 min of exposure, suggesting that efflux pumps are an immediate response to biocide treatment but may not be maintained in the long term. This could be due to the high energy and co-substrate demands for the use of efflux pumps.

The aim of the RNA-Seq work was to screen for novel QAC-resistance genes. Appendix A present a list of novel gene targets, as these genes had the highest observable fold change and greatest Z score, and thus, were most likely to play a role in the high levels of resistance of *Serratia* sp. HRI. Interestingly, a large proportion of the most significantly differentially expressed genes are uncharacterised or classified as hypothetical currently, suggesting a large proportion of the genes involved in QAC resistance remain uncharacterised. Additionally, many mobile genetic elements or mobile-associated genes were found to be significantly differentially expressed, confirming the high dissemination potential of QAC resistance and the notion that *Serratia* species have an unusually high genome plasticity [22].

Previous work has highlighted that the greater the fold change in the differential expression in RNA-Seq results, the greater the reliability of the result [22,23]. Everaert and co-workers [23] compared the expression of genes evaluated with both RNA-Seq and qPCR and found that the differential expression values of only 1.8% of genes significantly differed in a specific sub-set of genes. The 1.8% of non-comparable genes were all found to be smaller, with fewer exons (in eukaryotic studies), and differentially expressed with less than a two-fold change [22,23]. In light of this, only the genes with the greatest fold changes and highest Z scores were considered, as these would be the most reliable results.

The uncharacterised MFS efflux pump gene introduced in this study was found to be up-regulated at all four timepoints of BC exposure in the RNA-Seq data. Furthermore, the real-time PCR data revealed that this gene was up-regulated for all three QAC biocides used in this study (benzalkonium chloride, DDAC and Virukill^TM^). Interestingly, different levels of up-regulation were noted among the three generations of QAC biocides, indicating slight changes in the expression levels in response to different biocides in the same group. This corresponds to work performed in *Escherichia coli*, where the gene expression profiles differed depending on the type of biocide used [21]. Therefore, this suggests that not only can gene expression differ between groups of biocides but expression profiles may differ between generations of biocides within the same group.

## 5. Conclusions

The up-regulation of the uncharacterised MFS efflux pump elucidated in this work, as confirmed by RNA-Seq and real-time PCR expression studies, suggests this to be a strong candidate for a novel QAC-resistance gene. Further analysis is recommended to confirm the effect of this efflux pump on the resistance phenotype and the extent of its role in antimicrobial susceptibility to not only QAC-biocides but antibiotics and other antimicrobials as well. This work showcases the power of “omics” tools for future work in screening and identifying novel antimicrobial resistance genes. The molecular characterisation of antimicrobial resistance is vital to the understanding of the epidemiology of resistance. Once resistance genes have been characterised, this information can be used in the surveillance of antimicrobial resistance and in the tracking of the mobility and transfer of resistance determinants, and it can elucidate how resistant populations develop.

## Figures and Tables

**Figure 1 microorganisms-12-01891-f001:**
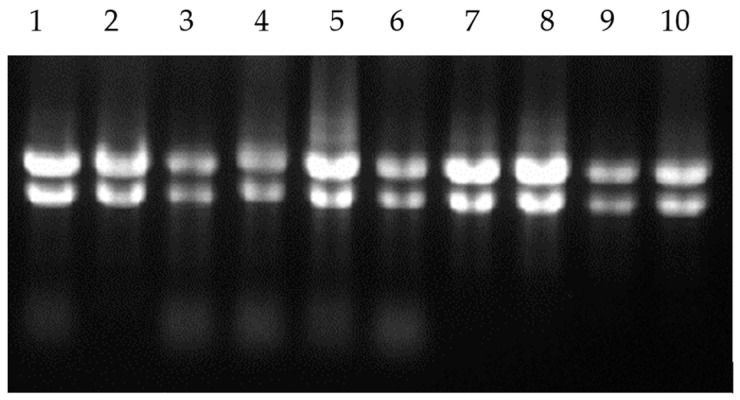
Total RNA extracts visualised on an agarose gel during the RNA extraction optimisation representing the 23s rRNA and 16s rRNA from each sample.

**Figure 2 microorganisms-12-01891-f002:**
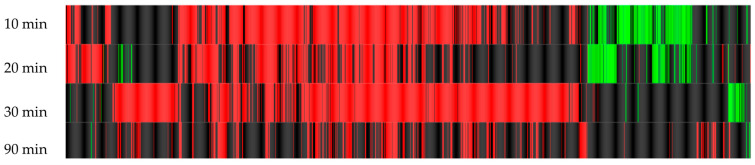
RNA-Seq differential expression heatmaps compared to the untreated control at four time points. Each line represents a gene: black indicates no differential expression; red indicates an up-regulation; green indicates down-regulation of a gene.

**Figure 3 microorganisms-12-01891-f003:**
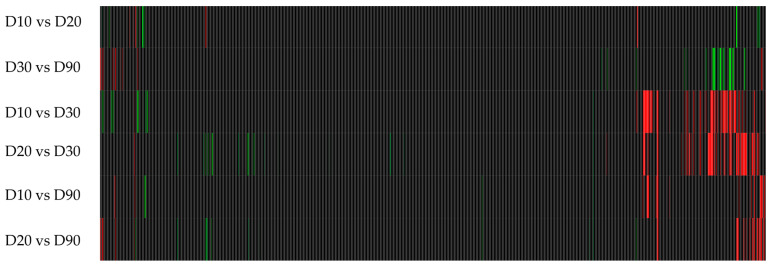
RNA-Seq differential expression heatmaps of the treated samples at different time points. Each line represents a gene: black indicates no differential expression; red indicates an up-regulation; green indicates down-regulation of a gene.

**Figure 4 microorganisms-12-01891-f004:**
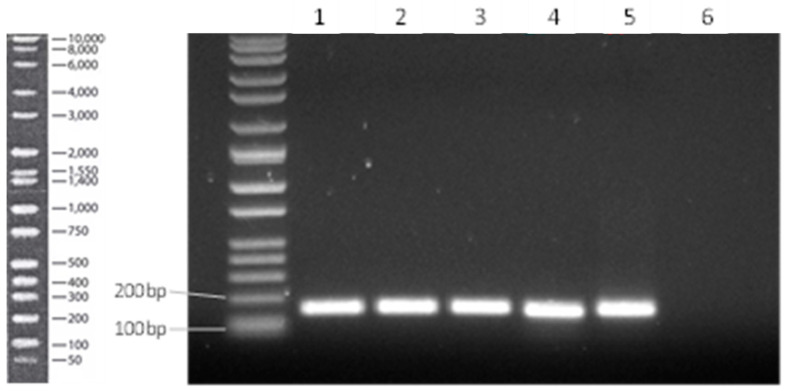
Successful amplification of the uncharacterised MFS efflux pump (UMFS1) target gene from the pooled cDNA biological replicates with the designed primers with no visible non-specific binding. Lane 1: untreated/control pooled sample; Lane 2: BC 20 pooled sample, Lane 3: BC90 pooled sample; Lane 4: DDAC pooled sample; Lane 5: Virukill^TM^ pooled sample; Lane 6: negative reverse transcriptase control.

**Figure 5 microorganisms-12-01891-f005:**
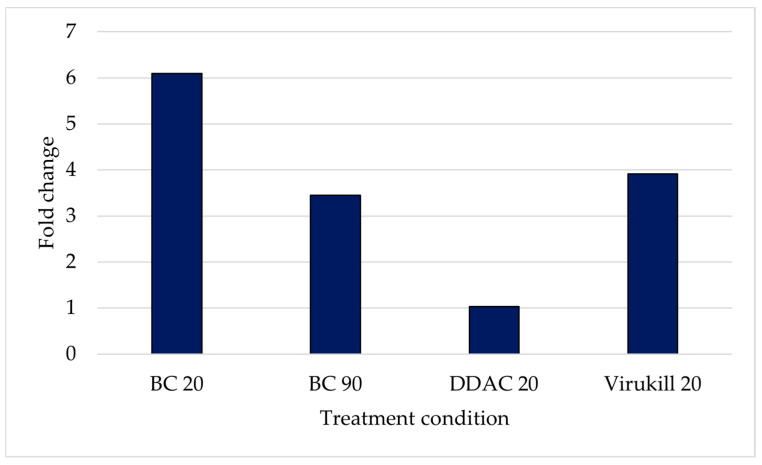
Calculated fold change in the gene expression of UMFS1 gene after exposure to three different QAC-based disinfectants (BC: benzalkonium chloride; DDAC: didecyldimethylammonium chloride; Virukill: QAC-based disinfectant), at either 20 or 90 min of contact time, compared to untreated samples evaluated by quantitative real-time PCR.

**Table 1 microorganisms-12-01891-t001:** Experimental conditions and abbreviated sample names used in this study.

Condition and Timepoint of RNA Extraction	RNA-Seq Sample Name Abbreviation
Untreated/Control 10 min	+10
Untreated/Control 20 min	+20
Untreated/Control 30 min	+30
Untreated/Control 90 min	+90
Benzalkonium chloride exposure for 10 min	D10
Benzalkonium chloride exposure for 20 min	D20
Benzalkonium chloride exposure for 30 min	D30
Benzalkonium chloride exposure for 90 min	D90

**Table 2 microorganisms-12-01891-t002:** Target gene details and primer sequences used in this study.

Uncharacterised MFS-Type Transporter	
BV-BRC Identifier	fig|2663241.3.peg.999
RefSeq Identifier	GF111_02590
atgagtgccttacacgccggcggcggcgcaaaaccctggctggcgactttcgctatcgccctgtccacctttaccgtcgtcacggcagaaatgctgcccgtcgggctgctgactccgatcgtcagtaccctgaatgcgacgatcggccgcgccggattactggtttccctgcccgccctgtttgccgcgctgttcgccccgctggtggtgctgggtgcccggcgcacggatcgccgcagcctgctgatcgcctttctgctgctgctgatcgccgccaacctgctggccgccgccgccacctccatggcgctgctgttcactgcgcgcatcctgctgggcttttgcatcggcggtatctgggccatcgcaggtgggttggcggaacgtctggtgccgccgacttccgtaggattggcgctgtccgtcatcttcagcggcgtagcggcagcctcggtgttcggcgtgccgctcggggtattcctcggcgaagcgctgggctggcgcatggcgtttctggccgtcgccgtcctcgcggcgctgaccctgctcctgctgctgtgcgtgctgccgccgctgccggtcacgcagacggtgagctggcgtttattcaccgcgcagcgtgcgaatcgccgactgctcctcgggctgctgctgacgtttctgctggtcgccggccactttatggcctacacctttgtccgcccgctgctgcagacggtcgccggcatcgagggccgctgggtggggccgctgctgttcgcctacggcgccgccggtatcgtcgggaatttcatcgccggccaggccgcggccaaacggctacgccgcaccctggcgctgatcgccctcgggctggcgctcgccgtcctgctgctgccgctgctgggccacgcgccgctgagcggcagcgccttcctcctgctgtggggcatcgcctacggcggcgtttccgtttcgctgatggcatggatgctcaaggcggcccccgatgcggtcgaggtcgcctcttcgctgtatatcgcgctgtttaatctggcgatttcctgcggttcactggcgggcgggctggtagtggacgccggaggcttgacgataaacggcgtgctgtccggcatcgtgctgctgctggcgttggcgatcctgatgcgaacccgcccacaggcgctgacaacggcggcgaaggccgattcgccgccgggttga	
Designed forward primer sequence	5′-TTCCGTTTCGCTGATGGC-3′
Designed reverse primer sequence	5′-TGAACCGCAGGAAATCGC-3′

**Table 3 microorganisms-12-01891-t003:** RNA samples for the RNA-Seq and qPCR experiments.

Sample	NanoDrop Conc. (ng/µL)	A_260/280_	A_260/230_	Qubit BR Conc. (ng/µL)	TapeStation RIN
+10	686	2.07	2.27	884	10.0
+20	290	2.19	1.23	398	9.7
+30	400.8	2.18	1.29	552	9.7
+90	638.9	2.15	1.28	876	10.0
D10	571	1.83	1.83	286	8.4
D20	2979	1.51	1.31	1080	9.6
D30	443	1.92	2.25	340	9.1
D90	1076.0	2.19	1.82	1300	10.0

**Table 4 microorganisms-12-01891-t004:** Number of genes with a DGE fold-change ≥ 3 and Z-score ≥ 3 compared to the control samples. The genes meeting these criteria were further assigned to four main systems found to be differentially expressed after exposure to BC.

Systems Most Affected in the Literature [21]	10 min	20 min	30 min	90 min
Motility	8	1	4	12
Membrane composition and transport	14	4	11	5
Metabolism adaptions	13	11	11	17
Stress response	7	2	3	3
Uncharacterised	26	18	19	27
Total	68	36	48	64

**Table 5 microorganisms-12-01891-t005:** N sample concentrations after RNA conversion to cDNA for 5 conditions in triplicate.

Treatment	Sample Name	Concentration (ng/µL)
Untreated/Control	+A	93.7
	+B	98.1
	+C	96.8
Benzalkonium chloride (20 min)	B2A	91.9
	B2B	75.4
	B2C	66.5
Benzalkonium chloride (90 min)	B9A	50
	B9B	37.3
	B9C	58.2
DDAC (20 min)	DA	146
	DB	95.8
	DC	120
Virukill^TM^ (20 min)	VA	90.8
	VB	110
	VC	52.8
Negative Reverse Transcriptase	-RT	2 (No change)

**Table 6 microorganisms-12-01891-t006:** Characteristics of the standard curves generated for the target gene.

	Standard Curve 1	Standard Curve 2	Standard Curve 3
Slope	−3.584	−3.193	−3.246
Efficiency (%)	90%	106%	103%

**Table 7 microorganisms-12-01891-t007:** Comparison of the calculated fold change in the gene expression of the UMFS1 gene after BC treatments for the RNA-Seq and qPCR data.

Timepoint (min)	Fold Change in Expression RNA-Seq Data	Fold Change Expression qPCR Data
10	5.13279	N/A
20	5.44252	6.09
30	4.16596	N/A
90	4.26023	3.45

## Data Availability

The sequence data are available on request. The RNA-Seq data analysis methodology is freely and openly available at the Bacterial and Viral Bioinformatics Resource Center (BV-BRC) available at “https://www.bv-brc.org/ (accessed on 10 June 2023).”

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
