# Peer review of "Differential Expression Analysis Reveals Possible New Quaternary Ammonium Compound Resistance Gene in Highly Resistant Serratia sp. HRI"

_microorganisms, 2024, doi:10.3390/microorganisms12091891_

Round 1

Reviewer 1 Report

Comments and Suggestions for Authors

Dear authors

Thanks for your work and presentation.

However, some comments should be considered during revision;

  1. Some abbreviations without full definition (MFS in line 18, EPA in line 38, MIC line 71, E. coli in line 330, etc.).
  2. A hint about the importance of the mentioned bacterial strain (Serratia sp. HRI) either in humans or animals environment should be mentioned in the introduction.
  3. A hint about the uses of tested disinfectants should also mentioned in the introduction.
  4. More details about the origin of the used bacterial strain should be noted.
  5. A lot of details are present in the used methods. They could be replaced by references. In case of the presence of any modifications in the methods, they could be mentioned in brief.
  6. No references should be added in the results section (No. 20 in line 235).
  7. Lines 148 and 317, McClure and co-workers (2013)[18] and Everaert and co-workers (2017), the number of the reference should be mentioned instead of the year.
  8. The conclusion should be stated as a title and supported with some recommendations.

Best wishes.

Author Response

Reviewer comment: Some abbreviations without full definition (MFS in line 18, EPA in line 38, MIC line 71, E. coli in line 330, etc.).

Author response: Corrected- Abbreviations have been defined.

Reviewer comment: A hint about the importance of the mentioned bacterial strain (Serratia sp. HRI) either in humans or animals environment should be mentioned in the introduction.

Author response: Corrected- The significance of this species has been elaborated with the unique abilities of this strain.

Reviewer comment: A hint about the uses of tested disinfectants should also mentioned in the introduction.

Author response: Corrected- The uses of QAC disinfectants are added into the introduction.

Reviewer comment: More details about the origin of the used bacterial strain should be noted.

Author response: Corrected- More details on the origin of this isolate has been added.

Reviewer comment: A lot of details are present in the used methods. They could be replaced by references. In case of the presence of any modifications in the methods, they could be mentioned in brief.

Author response: Corrected- Methods have been adjusted and areas where methods have been excluded due to following manufacturers instructions have been highlighted.

Reviewer comment: No references should be added in the results section (No. 20 in line 235).

Author response: Corrected- Reference has been removed.

Reviewer comment: Lines 148 and 317, McClure and co-workers (2013)[18] and Everaert and co-workers (2017), the number of the reference should be mentioned instead of the year.

Author response: Corrected- References have been amended.

Reviewer comment: The conclusion should be stated as a title and supported with some recommendations.

Author response: Corrected- Conclusion title added and future recommendations clarified.

Reviewer 2 Report

Comments and Suggestions for Authors

The manuscript is devoted to the analysis of disinfectant-mediated expression in highly resistant Serratia sp. strain HRI.

Since Serratia is a clinically significant genus of bacteria, its study has undeniable practical benefits. The qac genes encoding efflux pumps are thoroughly studied compared to other disinfectant resistance genes, but represent an interesting target and a priority for further research.

The manuscript describes a detailed and meticulous work. One of the important and interesting surveys is a comparative analysis of RNA sequences of samples treated with benzalkonium chloride with different exposure times, as well as the subsequent conclusion that all multidrug and resistance efflux pumps were most differentially expressed within the first minutes of exposure. As a continuation and confirmation of this conclusion, it is reasonable to perform a similar test with other disinfectants used in this work and clinical practice.

In general, the manuscript contains a few examples of bad formatting and typos. Some of them are listed below:

Lines 39, 72, 148. Missing spaces before links.

Line 208. The caption for Table 3 is on page 5, and Table 3 itself is on page 6, making it difficult to understand the content.

Line 297. “in the short-term (Up to 30-minutes) response” probably a lowercase u was intended.

Appendix A. Table A5 is mentioned, but there is no Table A5 itself.

Author Response

All reviewer comments and recommendations have been implemented.

Reviewer comment: The manuscript is devoted to the analysis of disinfectant-mediated expression in highly resistant Serratia sp. strain HRI.

Since Serratia is a clinically significant genus of bacteria, its study has undeniable practical benefits. The qac genes encoding efflux pumps are thoroughly studied compared to other disinfectant resistance genes, but represent an interesting target and a priority for further research.

The manuscript describes a detailed and meticulous work. One of the important and interesting surveys is a comparative analysis of RNA sequences of samples treated with benzalkonium chloride with different exposure times, as well as the subsequent conclusion that all multidrug and resistance efflux pumps were most differentially expressed within the first minutes of exposure. As a continuation and confirmation of this conclusion, it is reasonable to perform a similar test with other disinfectants used in this work and clinical practice.

In general, the manuscript contains a few examples of bad formatting and typos. Some of them are listed below:

Lines 39, 72, 148. Missing spaces before links.

Author response: Corrected- Spaces before in-text references added.

Line 208. The caption for Table 3 is on page 5, and Table 3 itself is on page 6, making it difficult to understand the content.

Author response: Corrected- Spacing of manuscript after all revisions amended.

Line 297. “in the short-term (Up to 30-minutes) response” probably a lowercase u was intended.

Author response: Corrected.

Appendix A. Table A5 is mentioned, but there is no Table A5 itself.

Author response: Corrected- Appendix numbering revised.
